# Evaluating the Anesthetic and Physiologic Effects of Intramuscular and Intravenous Alfaxalone in Eastern Mud Turtles (*Kinosternon subrubrum*)

**DOI:** 10.3390/ani14030460

**Published:** 2024-01-31

**Authors:** Stephanie Zec, Mark A. Mitchell, Kelly Rockwell, Dana Lindemann

**Affiliations:** 1College of Veterinary Medicine, University of Illinois, 1008 W. Hazelwood Drive, Urbana, IL 61802, USAdana.lindemann@gmail.com (D.L.); 2Wildlife Conservation Society, Bronx Zoo, 2300 Southern Boulevard, Bronx, NY 10460, USA; 3School of Veterinary Medicine, Louisiana State University, Skip Bertman Drive, Baton Rouge, LA 70803, USA; 4Sea World, 7007 Sea Harbor Drive, Orlando, FL 32821, USA

**Keywords:** alfaxalone, sedation, chelonians, *Kinosternon subrubrum*, eastern mud turtle

## Abstract

**Simple Summary:**

Turtles and tortoises are routinely presented to veterinarians for medical and surgical treatment. In many of these cases, sedation is required to properly examine the animal and perform necessary diagnostic tests. Unfortunately, there is a dearth of evidence-based research available to guide veterinarians working with these animals. In addition, the available publications show that there are differences between species. Therefore, with nearly 350 species of chelonians in the world, it is important for us to evaluate protocols on a species-by-species basis. In this study, six eastern mud turtles (*Kinosternon subrubrum*) were given 10 mg/kg alfaxalone via intramuscular (IM) and intravenous (IV) routes using a cross-over study design that allowed us to compare the results between and within turtles. As expected, the IV route led to a faster induction and recovery. This study demonstrated that alfaxalone 10 mg/kg IV or IM can be used to provide safe and effective sedation in eastern mud turtles.

**Abstract:**

Current sedation protocols for chelonians can pose a challenge to clinicians because of prolonged induction and recovery times, difficulties in gaining venous access, and natural species variation. This study evaluated the sedative and physiologic effects of intramuscular (IM) and intravenous (IV) alfaxalone in six wild-caught adult eastern mud turtles (*Kinosternon subrubrum*). The turtles received alfaxalone 10 mg/kg IM and IV in a randomized cross-over design. A 10-day washout period occurred between trials. Baseline parameters (heart rate, respiratory rate, temperature, and reflexes) were assessed prior to injection and every 5 min post-injection until recovery. Three venous blood gas samples were also collected and analyzed over the course of each trial (baseline, induction, and recovery). Intravenous alfaxalone resulted in a significantly faster induction (*p* = 0.016; median: 1.5 min, 25–75%: 1–7.5, minimum–maximum: 1–21) and a shorter total sedation time (*p* = 0.041; median: 52 min, 25–75%: 34.5–62.5, minimum–maximum: 33–87) when compared with IM alfaxalone (induction, median: 20 min, 25–75%: 15–22.5, minimum–maximum: 15–25; total, median: 70 min, 25–75%: 65–82.5, minimum–maximum: 65–90). Blood gas and physiologic parameters were not significantly different between groups; however, the pH (*p* = 0.009) and glucose (*p* = 0.0001) significantly increased, and partial pressure of carbon dioxide (*p* = 0.024) significantly decreased over time. This study demonstrated that alfaxalone 10 mg/kg IV or IM can be used to provide safe and effective sedation in eastern mud turtles.

## 1. Introduction

Current sedation and anesthetic protocols for chelonians can pose a challenge to clinicians because of prolonged induction and recovery times, poor-quality sedation or anesthesia, difficulties in gaining venous access, and natural species variation [1,2,3,4,5,6,7,8,9,10]. To overcome these issues, it is important to develop evidence-based methods to safely and effectively sedate and anesthetize these animals.

Alfaxalone is a neurosteroid anesthetic that has been approved by the U.S. Food and Drug Agency for use in veterinary medicine. While targeted as an anesthetic for dogs and cats [11,12,13,14], it is also being used in exotic species [15,16,17,18,19,20,21,22,23]. Alfaxalone acts on GABA A receptors, leading to a decrease in neuronal stimulation and producing clinical sedation and hypnosis. The current formulation of alfaxalone is water-soluble due to the 2-hydroxypropyl beta-cyclodextrin structure; this allows for subcutaneous (SC), intramuscular (IM), or intravenous (IV) administration [24]. The flexibility in the route of administration makes alfaxalone a viable sedative to consider for chelonians. 

To date, there have been only a few attempts to assess the value of alfaxalone as a sedative in chelonians, and the results have been variable [18,19,21,22,25,26]. In Macquarie River turtles (*Emydura macquarii*), alfaxalone 9 mg/kg IM did not produce sufficient sedation, whereas a dose of 10 mg/kg IM in Horsfield’s tortoises (*Agrionemys horsfieldii*) produced moderate sedation [19,21]. Red-eared sliders (*Trachemys scripta elegans*) administered alfaxalone 5 mg/kg IV could be intubated with an endotracheal tube soon after drug administration [25], whereas intubation was not possible in Macquarie River turtles at 9 mg/kg IV [21]. The results of these studies suggest that alfaxalone may be used as a sedative in chelonians, but that evidence should be generated on a species-by-species basis to ensure best success. Although IV and IM routes of administration have been compared in chelonians, to the authors’ knowledge, no studies have evaluated the physiologic effects of alfaxalone using blood gas analysis [21,26].

The purpose of this study was to determine the sedative and physiologic effects of IM and IV alfaxalone in eastern mud turtles (*Kinosternon subrubrum*). This freshwater species was selected because it is a bog species capable of torpor, which could affect the way it responds to apnea, bradycardia, or other physiologic changes associated with a sedation event. The specific hypotheses being evaluated were that (1) both IV and IM alfaxalone could be used to induce sedation in eastern mud turtles, (2) induction and recovery times would be faster in turtles given IV versus IM alfaxalone, and (3) acidemia would occur from resultant apnea at a deep anesthetic plane when the second blood sample was performed but would return to baseline following anesthetic recovery. The first and second hypotheses were both directional one-tailed hypotheses, while the third hypothesis was a non-directional hypothesis because of the three levels in the independent variable, time (baseline, sedation, recovery).

## 2. Materials and Methods

This study was conducted in accordance with the protocols established by the University of Illinois Institutional Animal Care and Use Committee (protocol #:15-228). Six wild-caught adult mud turtles were acquired from a commercial pet shop (Sailfin Pets, Champaign, IL, USA) for this randomized cross-over study. The turtles were acclimated for a minimum of 7 days prior to being recruited into the study. The turtles were housed under standard conditions for the species, and the husbandry was considered appropriate by the United States Department of Agriculture inspector. The sex ratio (one male, five females) was based on animal availability and determined using sexual dimorphic characteristics for this species (head size, vent positioning, and presence of a phallus). The mean body weight of the turtles was 255 g (SD: 50.5 g, min–max: 165 to 315 g). This project was conducted during the natural summer for this temperate species.

The turtles were randomly assigned to two groups using a random number generator (www.random.org, accessed on 15 October 2015): group 1, IV alfaxalone first, IM alfaxalone second; group 2, IM alfaxalone first, IV alfaxalone second. The turtles were fasted for 12 h prior to the sedation trials. On the day of the trial, the turtles were transported via automobile from the pet store to the University of Illinois, College of Veterinary Medicine (Urbana, IL, USA). Turtles were temporarily housed on the day of the experiment in an incubator with an internal temperature of 27.2 °C (81 °F). Once acclimated, approximately 1 h later, each turtle was examined. No abnormalities were noted during the physical examinations.

Prior to sedating the turtles, baseline readings for heart rate (HR), respiratory rate (RR), carapace surface temperature, weight, and reflexes were recorded. A crystal Doppler (Parks Medical Electronics Inc., Aloha, OR, USA) was used to measure HR, and gular movements were used to determine RR. The surface temperature of the carapace was collected with an infrared thermometer (Fluke, Everett, WA, USA). Superficial and deep nociception, the palpebral reflex, and the righting reflex were evaluated. The baseline reaction to noxious stimuli was evaluated using padded hemostats on the skin of the vent, forelimbs, and hindlimbs. Deep pain was also tested during the procedure using padded hemostats on the distal phalanx of the left forelimb. Reactions to the reflexes were recorded using a rank scale: 1—immediate response, 2—mild response, 3—no response. Baseline venous blood gases were also collected at this time. The blood samples were collected from the jugular vein using a 1 mL syringe fastened to a 25-gauge needle. No more than 0.2 mL of blood was collected for any of the blood samples. The samples were processed within two minutes of collection. Packed cell volumes (PCV) were measured using 40 mm heparinized microhematocrit tubes (LW Scientific, Lawrenceville, GA, USA) and centrifuged at 12,000 rpm/7500× *g* in a ZipCombo centrifuge (LW Scientific, Lawrenceville, GA, USA) for three minutes. The PCV were measured using a microhematocrit capillary tube reading card (Veterinary Information Network, Davis, CA, USA) in duplicate and the values were averaged. The plasma was removed from the microhematocrit tubes and placed on a clinical refractometer (Jorgenson Laboratories, LLC, Loveland, CO, USA) to measure the total solids (TS). A CG 8+ I-Stat cartridge (Abaxis Inc., Union City, CA, USA) and I-Stat 1 machine (Abaxis, Inc. Union City, CA, USA) were used to measure hemoglobin, glucose, sodium, potassium, ionized calcium, pH, partial pressure of carbon dioxide (pCO_2_), partial pressure of oxygen (pO2), bicarbonate (HCO_3_), and total carbon dioxide (TCO_2_). The surface body temperature of each turtle at the time of blood collection was entered into the I-Stat machine to adjust the algorithm for calculating the different parameters. Hematocrit and oxygen saturation were measured in the CG8+ cartridges but were not included because these values are unreliable for reptiles [27,28].

After baseline readings were collected, 10 mg/kg alfaxalone was administered to the group 1 turtles in the right jugular vein, while group 2 turtles received 10 mg/kg alfaxalone in the right biceps brachii muscles. The turtles were manually restrained in lateral recumbency by a single handler, and a second individual administered the alfaxalone. The IV and IM injections were delivered via a 25-gauge needle fastened to a 1 mL syringe. Once the needle was inserted into the jugular vein, the plunger was withdrawn to visualize blood in the hub/syringe prior to administering the drug over a 15 s span. The jugular vein was observed during the delivery of the alfaxalone to ensure the drug was not administered perivascularly. In one case, the subcarapacial sinus was used for the IV delivery of alfaxalone because it was not possible to access the jugular vein post-baseline blood sampling. The HR, RR, carapace temperature, palpebral reflex, righting reflex, and pain responses were recorded every 5 min until the turtles were recovered. Supplemental heat was provided during the anesthetic trial using a water-recirculating heating pad. Induction was considered to be from the point of drug administration to the time of full sedation, which was when the turtle had no withdrawal or righting reflexes (all reflexes = 3). A second blood sample was collected when the turtles achieved full sedation. Intubation was not attempted, as periods of apnea were brief. Recovery was determined to be the time from full sedation to the point when the reflexes returned to baseline values (all reflexes = 1). A final blood sample was collected at recovery. The total blood volume required for all 3 blood samples was <1% of the turtles’ body weights. Once fully recovered, the turtles were returned to the pet shop.

After a 10-day washout period, the turtles were transported to the University of Illinois for the second trial. Turtles from group 1 received 10 mg/kg alfaxalone IM, and turtles in group 2 received the same dose IV. All animals were processed as previously described. All of the turtles recovered uneventfully in both trials. At the end of the study, the turtles were returned to the pet shop.

The data were evaluated using the Shapiro–Wilk test, skewness, kurtosis, and q-q plots. Data that met the assumption of normality are reported by the mean, standard deviation (SD), and minimum–maximum (min–max) values, while non-normally distributed data are reported by the median, 25–75 percentiles (%), and min–max values. Linear mixed models were used to analyze the data, with turtle serving as the random variable and route of injection (IV or IM), order of injection (first trial, second trial), and time serving as fixed factors; sex was not included because there was only a single male. Interaction terms were also included in the initial models. The Akaike information criterion was used to help assess model fit. The Shapiro–Wilk test and histograms were used to evaluate the distributions of the residuals. The data were also assessed for outliers visually on box plots and using the Dixon test. Scatterplots and q-q plots were used to evaluate the homoscedasticity of the residuals. Post hoc analysis was performed with a Bonferroni test. Significance for statistical comparisons was set at alpha ≤ 0.05. Post hoc power analysis was performed for *p* = 0.06–0.10. SPSS 24.0 (IBM statistics, Armonk, NY, USA) was used to analyze this dataset.

## 3. Results

Intravenous alfaxalone resulted in significantly faster induction to full sedation (F = 6.6, *p* = 0.016) than IM alfaxalone, with median induction times being 13.3 times faster via the IV route (Table 1); there was no difference by order of injection (F = 1.573, *p* = 0.265). There was a single outlier (turtle #4) in the IV induction group that impacted the overall induction time. The induction time for turtle #4 following IV alfaxalone was 21 min, while the induction time for the other 5 turtles was 1–3 min. There was also a significant difference in total sedation time (F = 3.96, *p* = 0.041), with the IM route being 1.35 times longer than the IV route (Table 2); order of injection did not impact total sedation time (F = 1.914, *p* = 0.204). There was no significant difference in the time to recovery between the two dosing routes (F = 1.99, *p* = 0.216) or order of injection (F = 1.54, *p* = 0.269) (Table 2).

There were significant differences in surface temperature over time (F = 4.86, *p* = 0.018) but no significant differences related to the dosing routes (F = 1.59, *p* = 0.227) or order of injection (F = 0.029, *p* = 0.869). Significant differences over time were noted between baseline and induction (*p* = 0.016) and baseline and recovery (*p* = 0.002); there was no difference in surface temperature between induction and recovery (*p* = 0.393) (Table 2). There were no statistically significant changes in HR over time (F = 0.093, *p* = 0.772), order of injection (F = 0.01, *p* = 0.919), or between dosing routes (F = 1.12, *p* = 0.376). Because there were no differences in HR by group, order, or over time, the data for each turtle were compiled to provide baseline HR data for this species (Table 3). Respiratory rates also did not differ significantly over time (F = 0.180, *p* = 0.832), order of injection (F = 2.4, *p* = 0.119), or between the dosing routes (F = 0.06, *p* = 0.942) (Table 3).

There was a significant difference in glucose over time (F = 15.53, *p* = 0.0001) but no significant differences related to the route of injection (F = 0.806, *p* = 0.458) or order (F = 0.077, *p* = 0.789). Blood glucose was significantly higher at recovery when compared with baseline (*p* = 0.0001) and induction (*p* = 0.002) (Table 4). There was no significant difference between baseline and induction (*p* = 0.083), but the comparison did approach significance. The power for this comparison was low (power = 0.183). There was a significant difference in pH over time (F = 6.79, *p* = 0.009) but no significant differences in the route of injection (F = 1.21, *p* = 0.352) or order (F = 3.913, *p* = 0.076). Blood pH was significantly higher at baseline (*p* = 0.006) and recovery (*p* = 0.0001) when compared with induction (Table 4). There was no significant difference between baseline and recovery (*p* = 0.376). pCO2 was significantly different over time (F = 4.331, *p* = 0.024) but not by route of injection (F = 1.827, *p* = 0.181) or order (F = 0.021, *p* = 0.885). pCO_2_ was significantly lower at recovery than baseline (*p* = 0.029); there were no significant differences between baseline and induction (*p* = 0.323) or induction and recovery (*p* = 0.076), although the difference between induction and recovery approached significance. The power for this second comparison was low (0.283). There were no significant differences in PCV (route: F = 0.675, *p* = 0.518; order: F = 0.172, *p* = 0.689; time: F = 0.730, *p* = 0.492), total solids (route: F = 3.01, *p* = 0.10; order: F = 0.039, *p* = 0.848; time: F = 0.657, *p* = 0.520), hemoglobin (route: F = 2.48, *p* = 0.11; order: F = 0.078, *p* = 0.789 ; time: F = 1.01, *p* = 0.38), ionized calcium (route: F = 0.61, *p* = 0.551; order: F = 0.841, *p* = 0.367; time: F = 0.318, *p* = 0.73), sodium (route: F = 0.0.713, *p* = 0.499; order: F = 0.199, *p* = 0.659; time: F = 0.273, *p* = 0.763), potassium (route: F = 2.86, *p* = 0.107; order: F = 0.915, *p* = 0.371; time: F = 0.71, *p* = 0.55), bicarbonate (route: F = 2.049, *p* = 0.149; order: F = 4.34, *p* = 0.052; time: F = 0.107, *p* = 0.899), base excess (route: F = 1.12, *p* = 0.341; order: F = 0.003, *p* = 0.959; time: F = 0.824, *p* = 0.450), or pO_2_ (route: F = 0.122, *p* = 0.885; order: F = 2.505 *p* = 0.151; time: F = 1.703, *p* = 0.202) by time or group.

## 4. Discussion

Alfaxalone dosed at 10 mg/kg IV and IM achieved a rapid and reliable plane of sedation in mud turtles. IV induction and total anesthesia times were significantly faster than IM induction, which was not unexpected. IV induction is the preferred method when attempting to induce any species, especially if time and efficiency are important; however, IV access is not always possible in chelonians, so additional routes, such as IM injection, are necessary. Subcutaneous routes should also be further investigated, as they have proved valuable in leopard geckos (*Eublepharis macularius*) and would be expected to induce less pain on injection [29]. The total duration of anesthesia found in the current study was comparable to a study in another temperate North American species of turtle, red-eared sliders [18,22,25]. These comparable findings should provide veterinarians with some confidence in how much time they will have when planning procedures for these species of turtles. IM injections of alfaxalone did achieve a deep level of sedation in the turtles in this study, which is different from the minimal sedation reported after alfaxalone dosing at 9 mg/kg IM in Macquarie River turtles (*Emydura macquarii*) [21]. It is not unexpected to see variations in responses to anesthetics across reptile species, and thus we should not make generalizations across taxa. We attempted to minimize any variance in delivery by using a single IM site for injection, as blood flow may vary between muscle sites.

In this study, the jugular vein was used for vascular access because of concerns about using the subcarapacial sinus; however, there was a single study subject that did require IV administration into the subcarapacial sinus because jugular venipuncture was not feasible. Though more easily accessible, the subcarapacial sinus was avoided by the authors due to concerns of intra-lymphatic or sub-meningeal drug administration. Propofol administered via the intra-lymphatic route in red-eared sliders resulted in an unpredictable, delayed, or light plane of anesthesia [30]. When propofol was accidentally administered into the meningeal space of a gopher tortoise (*Gopherus polyphemus*), severe bradycardia and a deep, long-duration anesthesia occurred [31]. In the current study, there were no side effects noted in the turtle given alfaxalone into the subcarapacial sinus, and this turtle was not the outlier regarding the IV induction time. While we did not encounter a complication with this animal, the authors strongly recommend against giving IV injections into the subcarapacial sinus unless they are image-guided (e.g., fluoroscopy).

In one turtle in the IV induction group (#4), there was an unpredictable pattern of sedation. The authors suspect that this was due to the visible hematoma that formed during drug delivery. This case did result in the longest sedation of all the IV protocols. While it did not affect the overall results, it is important for veterinarians to recognize that delayed responses may be encountered with the IV route with hematoma formation.

Due to the small size of the turtles, frequent blood sampling was challenging. Additionally, it was subjectively observed that obtaining the second blood sample was more difficult in the IV experiments. However, this was not attributed to hematoma formation, as the third blood sample was easily obtained. The challenges associated with the second IV sampling period may have been attributed to the effects of the alfaxalone on the turtles’ blood pressures. In dogs, alfaxalone administered IV at doses of 6 and 20 mg/kg led to a dose-dependent decrease in arterial systolic, diastolic, and mean blood pressures [12]. In cats, an alfaxalone dose of 10 mg/kg IM resulted in significant hypotension that was not observed at lower doses [14]. Bullfrogs (*Lithobates catesbeianus*) administered IV alfaxalone demonstrated a longer-acting significant decrease in mean arterial blood pressure compared with bullfrogs given IM alfaxalone [32]. These three examples suggest that the dose and IV route used in the current study could have had an effect on the increased challenges the authors noted when collecting blood samples. Direct blood pressure monitoring was not feasible in this study due to the invasive nature of the technique and small size of the turtles, and indirect methods for measuring blood pressure in chelonians have not been validated. It is important that we develop methods for measuring blood pressure in chelonians so that we can better monitor and manage them during these anesthetic events. It was also interesting to note that HR and RR were not significantly different in these experimental animals, although variability did certainly occur (note minimum–maximum values in Table 3). While not significant, the turtles did demonstrate some clinical physiologic changes in these measurements during the anesthetic events.

All sedation events resulted in the loss of nociceptive and righting reflexes, and intubation was possible in all of the animals regardless of the route of administration. However, unlike in other experiments, a loss and return of reflexes in a cranial-to-caudal direction was not observed [22], although cervical tone and control of the head were usually the last responses to return.

There were significant differences in venous pH and pCO_2_ over time. These changes were attributed to the brief periods in which the RR decreased during the trials. Again, while not significant, reductions in RR were noted that correlated with these physiologic changes. The differences in pCO_2_ did not follow the same significant changes noted with pH, but this was attributed to the variability in this continuous data type and the small sample size as noted by the low statistical power. Respiratory effects in dogs administered alfaxalone have been found to be dose-dependent, with little change in respiratory rate and PaCO_2_ at 1 mg/kg but more profound changes at 6 and 20 mg/kg [12]. In loggerhead sea turtles (*Caretta caretta*), respiratory acidemia only occurred at alfaxalone doses of 10 mg/kg IV [26]. In these turtles, a higher dose may have likewise demonstrated a more consistent reduction in RR, as well as altered the pCO_2_ accordingly. It is important for veterinarians to recognize that these changes occur and that using higher or lower doses may lead to more or less profound effects, respectively. Blood glucose was also found to increase over time, but this finding was attributed to the physiologic stress response associated with handling. All of the turtles demonstrated normal behaviors following the trials, so it was assumed that this stress effect was transient.

There were several limitations in this study that should be addressed. First, the sample size was small. Small sample sizes are an unfortunate hallmark of these types of clinical studies with reptiles. While sample size could have impacted some of the comparisons because of the continuous nature of the data, such as the respiratory rate, the sample size did not affect our primary hypotheses, as we were able to positively prove all three hypotheses. Another limitation was the method for measuring body temperature. Our original intent was to measure cloacal/colonic temperatures, but we had an equipment malfunction at the start of the study. To overcome this, surface temperatures were measured. The turtles were purposely housed in an incubator with a stable temperature and provided supplemental heat to ensure a consistent body temperature. Body temperature measurement was important to adjust the algorithm for the pH. Because the findings were consistent with our third hypothesis, we do not believe this shortcoming impacted the overall results, but we would have preferred to measure cloacal temperatures. Finally, we used gular pumping to measure RR. It was challenging to see head or limb pumping in these turtles, even at baseline, so we elected to use a method of monitoring that we could consistently measure. Gular pumping is associated with olfaction in turtles but has also been used to assist with measuring RR. In this species, the variability noted may have impacted our results and led to the challenge in characterizing the changes over time for the pCO_2_.

## 5. Conclusions

Alfaxalone at 10 mg/kg via IV or IM routes resulted in adequate anesthesia for minor procedures in eastern mud turtles. IV administration did result in a more rapid induction and shorter total anesthesia time and therefore may be preferred when time is of the essence. However, because IV administration of a drug can be challenging in chelonians, the results of this study suggest IM administration in the biceps brachii is just as effective at producing reliable anesthesia.

## Figures and Tables

**Table 1 animals-14-00460-t001:** Induction, recovery, and total anesthesia times (in minutes) for mud turtles receiving IV and IM alfaxalone.

Anesthetic State	Route	Median	25–75%	Min–Max
*Induction*	IV	1.5 ^a^	1–7.5	1–21
	IM	20 ^a^	15–22.5	15–25
*Recovery*	IV	51	31.7–62.5	31–66
	IM	60	50–75	50–80
*Total anesthesia time*	IV	52 ^b^	34.5–62.5	33–87
	IM	70 ^b^	65–82.5	65–90

Statistically different: ^a^
*p* = 0.006; ^b^
*p* = 0.026.

**Table 2 animals-14-00460-t002:** Surface body temperatures in ^o^C (^o^F) over time in mud turtles anesthetized with alfaxalone.

Anesthetic State	Media	25–75%	Min–Max
*Baseline*	26.1 ^a,b^	25.1–26.5	24.2–30
	(79)	(77.3–79.8)	(75.5–86)
*Induction*	28.3 ^a^	27.8–30.2	26.1–31.4
	(83)	(82.1–86.4)	(79–88.5)
*Recovery*	28.9 ^b^	28.2–30.7	26.7–32.8
	(84)	(82.7–87.2)	(80–91)

Statistically different: ^a^
*p* = 0.016; ^b^
*p* = 0.002.

**Table 3 animals-14-00460-t003:** Heart and respiratory rates of 6 mud turtles enrolled in this study.

Parameter	Turtle	Mean	SD	Min–Max
*Heart rates*	1	40	7.6	28–48
	2	51.3	12.2	36–68
	3	48	18.4	28–80
	4	48.7	13.7	32–68
	5	52	5.6	44–60
	6	49.3	7.9	36–60
*Respiratory Rates*	1	20 *	20–41 **	20–48
	2	20 *	8–23 **	4–24
	3	20 *	13–34 **	6–48
	4	40 *	22–60 **	20–68
	5	58 *	17–78 **	8–80
	6	18 *	4.5–39 **	2–44

* median; ** 25–75%.

**Table 4 animals-14-00460-t004:** Blood gases found to be different over time in mud turtles anesthetized with alfaxalone.

Parameter	Anesthetic State	Mean	SD	Min–Max
*Glucose* (mg/dL)	Baseline	51.5 ^a^	9.34	39–66
	Induction	69.9 ^b^	13.1	39–78
	Recovery	87.5 ^a,b^	19.4	67–122
*pH*	Baseline	7.38 ^c^	0.11	7.23–7.62
	Induction	7.27 ^c,d^	0.09	7.12–7.45
	Recovery	7.42 ^d^	0.1	7.23–7.59
*pCO2* (%)	Baseline	45.2 ^e^	8.7	32–62.6
	Induction	51.6	9.98	32.5–69.8
	Recovery	38.6 ^e^	6.48	26.2–47.8

^a^ *p* = 0.0001; ^b^ *p* = 0.002; ^c^ *p* = 0.006; ^d^ *p* = 0.0001; ^e^ *p* = 0.024.

## Data Availability

Data are available to those who request it through the corresponding author: mmitchell@lsu.edu.

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
