# Peer review of "Evaluating the Anesthetic and Physiologic Effects of Intramuscular and Intravenous Alfaxalone in Eastern Mud Turtles (Kinosternon subrubrum)"

_animals, 2024, doi:10.3390/ani14030460_

Round 1

Reviewer 1 Report

Comments and Suggestions for Authors

To the Authors,

Thank-you for submitting this article for review, it is an important study with significant clinical applications. There are a few minor points, particularly associated with the methodology that need to be clarified before it is suitable for publication.

Line 60 (and throughout document): River should have a capital “R” as Macquarie River is the name of a place.

Line 65: Reference 25 is not relevant for Macquarie River turtles, should this be a general turtle reference?

Line 116: How were PCV and TS determined? Was this done manually or using the I-Stat as this method likely underestimates the haematocrit. Similarly sO2% is unreliable in the iStat in reptiles as it is calculated from the PCV.

See:

Muñoz-Pérez JP, Lewbart GA, Hirschfeld M et al. Blood gases, biochemistry and haematology of Galapagos hawksbill turtles (Eretmochelys imbricata). Conserv Physiol 2017;5:1–9.https://doi.org/10.1093/conphys/cox028.

Lewbart GA, Hirschfeld M, Denkinger J et al. Blood gases, biochemistry, and hematology of Galapagos green turtles (Chelonia mydas). PLoS One 2014;9: e96487. https://doi.org/10.1371/journal.pone.0096487

Wolf KN, Harms CA, Beasley JF Evaluation of five clinical chemistry analyzers for use in health assessment in sea turtles. JAVMA 2008;233:470–475.

Line 120: Why did you elect to take surface temperature and not cloacal temperature? Can you provide evidence that they are equivalent otherwise it calls into question the accuracy of your temperaturecorrections.

Line 124: How did you ensure that medication was administered IV? Did you use a catheter? Did you draw back on the syringe to see a flash of blood? Monitor for swelling around injection site for possible extravascular administration?

Lines 230-242: Why is this the first time this has been mentioned? You need to include this information in the MM section.

Lines 265-269: You didn’t notice any induction apnoea? This is almost universal for all injectable anaesthetics but especially for alfaxalone in a range of species. This may be evidence by the increase in pCO2 in your induction samples.

Line 275: I’m not convinced your pCO2 changes are correlated to RR changes as you have said that there were no significant differences across time points for this parameter. It is possible that your method of monitoring RR was inaccurate, i.e. observation and this has led to the observations here.

Author Response

Please see attached document (my laptop wont allow me to paste here, sorry). 

Reviewer 2 Report

Comments and Suggestions for Authors

The paper submitted by the authors described different ways of alfaxalone administration in the Mud turtles. The manuscript is well written and quite easy to read, providing interesting information about the metabolic response in these animals to this substance. Before acceptance, I only have some minor considerations.

In the introduction section, please delete all this information. Here, you only should name the  drug properties but not the brand or country.

In the material and methods section, I am not quite sure about the protocol number. It seems that the authors only performed this procedure based on  a regular protocol but not a specific one.

How you restrain the animal to apply the alfaxalone , please add this information.

Total sedation time, please add the information  as you did with the other parameters.

Line 267, is it Table 2? Please correct.

in the conclusion, please add the specific muscle.

Author Response

Please see attached document (my laptop wont let me past, sorry)
